# SiC MOSFET with Integrated SBD Device Performance Prediction Method Based on Neural Network

**DOI:** 10.3390/mi16010055

**Published:** 2024-12-31

**Authors:** Xiping Niu, Ling Sang, Xiaoling Duan, Shijie Gu, Peng Zhao, Tao Zhu, Kaixuan Xu, Yawei He, Zheyang Li, Jincheng Zhang, Rui Jin

**Affiliations:** 1Beijing Institute of Smart Energy, Beijing 102209, China; niuxiping@bise.hrl.ac.cn (X.N.); xigaoyn@163.com (L.S.); zhutao@bise.hrl.ac.cn (T.Z.); xukaixuan@bise.hrl.ac.cn (K.X.); heyawei@bise.hrl.ac.cn (Y.H.); lizheyang@bise.hrl.ac.cn (Z.L.); 2State Key Laboratory of Wide-Bandgap Semiconductor Devices and Integrated Technology, Xidian University, Xi’an 710071, China; duanxiaoling@xidian.edu.cn (X.D.); 23211214969@stu.xidian.edu.cn (S.G.); 23211215174@stu.xidian.edu.cn (P.Z.); jchzhang@xidian.edu.cn (J.Z.)

**Keywords:** SiC MOSFET, SBD, neural networks

## Abstract

The SiC MOSFET with an integrated SBD (SBD-MOSFET) exhibits excellent performance in power electronics. However, the static and dynamic characteristics of this device are influenced by a multitude of parameters, and traditional TCAD simulation methods are often characterized by their complexity. Due to the increasing research on neural networks in recent years, such as the application of neural networks to the prediction of GaN JBS and Finfet devices, this paper considers the application of neural networks to the performance prediction of SiC MOSFET devices with an integrated SBD. This study introduces a novel approach utilizing neural network machine learning to predict the static and dynamic characteristics of the SBD-MOSFET. In this research, SBD-MOSFET devices are modeled and simulated using Sentaurus TCAD(2017) software, resulting in the generation of 625 sets of device structure and sample data, which serve as the sample set for the neural network. These input variables are then fed into the neural network for prediction. The findings indicate that the mean square error (MSE) values for the threshold voltage (Vth), breakdown voltage (BV), specific on-resistance (R_on_), and total switching power dissipation (E) are 0.0051, 0.0031, 0.0065, and 0.0220, respectively, demonstrating a high degree of accuracy in the predicted values. Meanwhile, in the comparison of convolutional neural networks and machine learning, the CNN accuracy is much higher than the machine learning methods. This method of predicting device performance via neural networks offers a rapid means of designing SBD-MOSFETs with specified performance targets, thereby presenting significant advantages in accelerating research on SBD-MOSFET performance prediction.

## 1. Introduction

Silicon carbide (SiC) has been widely studied because of its advantages of high thermal conductivity, high critical breakdown electric field, and strong radiation resistance, which have the potential to be better than other semiconductors in high-frequency electronics, high-power devices, high-temperature electronics, and thermal sensors [1,2,3,4]. The superior material and electrical properties of SiC attribute render SiC MOSFETs particularly advantageous in applications such as electric vehicles [5], electrical machines, and various industrial sectors [6]. However, the substantial bandgap width results in a phenomenon known as bipolar degradation for the body diode of SiC MOSFETs [7]. Typically, the incorporation of an external reverse-parallel SiC Schottky diode (SBD) in SiC MOSFETs effectively alleviates the bipolar degradation and contributes to a reduction in reverse recovery charge and minimizes dead time losses [8,9,10]. Notably, the reverse recovery charge is temperature-independent, allowing for sustained low switching losses even at elevated temperatures [11,12,13]. And SiC Schottky diodes are often integrated directly into SiC MOSFET cells [14,15,16]. SiC power devices have become excellent candidates for high-power-density and high-efficiency applications under harsh temperature conditions [17].

Traditional methods of experimental simulation and testing, including the use of TCAD(2017) tools like Silvaco and Sentaurus, are often protracted and face challenges such as non-convergence. The modeling and simulation process consumes a lot of resources and time, so how to improve the simulation efficiency has become the focus of research. With rapid advancements in neural network technology, there is an emerging potential to utilize these networks for predicting the structure and performance of devices and materials. Machine learning has great advantages in dealing with complex parameter relationships, and it has also been widely used in the field of microelectronic devices in recent years [18,19,20]; for example, the I–V characteristics, threshold voltage, and other electrical characteristics of HEMT devices have been reported by machine learning-based prediction methods [21,22,23]. In this paper, a neural network prediction based on SBD-MOSFET is proposed. Given the multitude of influencing parameters and the complex mechanisms affecting the performance of SBD-MOSFET devices, it is imperative to optimize the design of the neural network to enhance its accuracy. In neural network prediction, the physical process is not involved, and the actual physical process cannot be characterized, and the TCAD simulation results can only be extrapolated, which has certain limitations, but we restore the reality as much as possible in the TCAD simulation and add sufficient physical field effects to ensure the accuracy of the data obtained by the device simulation. On the basis of adding the actual physical model, the neural network is further optimized, which will improve the accuracy of the neural network as much as possible under the limitations of neural network prediction [24].

In this study, we model and simulate the structure of SBD-MOSFET devices via Sentaurus TCAD software, resulting in the generation of 625 distinct device structures and corresponding sample data, which serve as the sample set. In the context of SBD-MOSFET devices, critical parameters such as threshold voltage significantly influence device performance. Research has indicated that these essential parameters are influenced by the various structural characteristics of the device. The input parameters considered in this research include P-well doping, the width of the JFET region, P-well depth, and Wsbd, while the output parameters encompass threshold voltage, specific on-resistance, breakdown voltage, and switching power dissipation. These output parameters effectively characterize device performance and are influenced by various input parameters [25,26], thereby enabling the prediction of device performance through neural networks. Following the accumulation of sample data via TCAD Sentaurus, the data will undergo training, and the accuracy of the predictive outcomes will be evaluated using accuracy metrics and mean square error (MSE).

## 2. TCAD Modeling and Simulation

### 2.1. Model Building

This paper’s device simulation and sample data accumulation are based on TCAD Sentaurus. Figure 1 shows the device model of SBD-MOSFET. The device structure includes the gate region, N+ region, P+ region, Pwell region, N-drift layer, N+ substrate, and Schottky-contact SBD structure. The specific device parameters are shown in Table 1.

To accurately replicate the forward and reverse characteristics of the device, a diverse array of physical models is employed in this experiment. These models include avalanche ionization, high field mobility, surface vertical electric field mobility, doping dependence, interfacial charge dynamics, bandgap narrowing, anisotropic effects, incomplete ionization, and Auger recombination.

As one of the basic characteristics of MOSFET devices, the threshold voltage affects the on-resistance and switching speed of the device, so in order to control these performances to give full play to the significant advantages of the device in power applications, the optimization of the threshold voltage has become one of the focuses of device design. In addition, as a power device designed to operate under high-voltage conditions, the SBD-MOSFET must consider the reliability of the Schottky barrier diode and the MOSFET when experiencing high-voltage blocking scenarios. Therefore, it is also necessary to obtain the device’s breakdown voltage through the BV graph output from the software and analyze it.

The IdVg diagram shows the relationship between the drain current and the gate voltage when the drain voltage is 10 V. As shown in Figure 2a, the threshold voltage, as derived from the curve, is measured at 6.101 V, which is higher than the value reported in the reference literature. This difference can be attributed to the fact that compared with the reference device [27], the actual modeling device changes the P-well doping mode, doping depth, and N+ doping mode, mainly to reduce the difficulty of simulation and enhance the convergence of simulation. Consequently, as shown in Figure 2c, this change also leads to a reduction in the device’s breakdown voltage, which is determined to be 1315 V from the BV graph, falling below the reference value. Furthermore, simulations are performed with V_GS_ set to 6 V, 9 V, 12 V, and 15 V, yielding the IdVd curve for the device. The resulting IdVd plot is shown in Figure 2b. Notably, the specific on-resistance is recorded at 4.0556 mΩ·cm^2^ when V_GS_ is 15 V. Although some structural parameters of the device have changed, the changing trend of its characteristic curve is basically consistent with the characteristic curve obtained in the literature. For example, compared with the threshold voltage of the reference paper, the threshold voltage increases by about 1.5 V, while the corresponding specific on-resistance decreases by about 50%. This is mainly caused by changing the surface concentration of P-well doping in the device. The deterioration of the threshold voltage is replaced by the optimization of characteristic resistance. These changes reflect the influence of modeling parameters on the device characteristics and also show the correctness of the device simulation [27].

### 2.2. Data Collection

The P-well region, the JFET region, and the SBD region greatly influence the static and dynamic characteristics of the SBD-MOSFET. To better represent the device, the input parameters examined in this study include P-well doping (P_wdopmax_), JFET region width (W_jfet_), P-well depth (P_wdepth_), and SBD width (W_sbd_) [25,26]. To ensure the accuracy of the predictive outcomes, it is essential to establish appropriate values for these input parameters. Figure 3 illustrates the impact of various input parameters on specific output parameters, facilitating an analysis of the acceptable value ranges for each input parameter. In order to ensure the correctness of the device parameters changed, we select the structural parameters set during the modeling of the device as the benchmark for pulling. The precise values for each input parameter are detailed in Table 2.

Figure 3 illustrates that as the doping concentration of the P-well increases, there is a corresponding decline in the breakdown voltage of the device, while the threshold voltage exhibits a gradual increase. Additionally, an increase in the width of the SBD results in a progressive rise in the specific on-resistance of the device. Conversely, an increase in the width of the JFET region leads to a reduction in the specific on-resistance. Furthermore, an increase in the depth of the P-well is associated with a decrease in the breakdown voltage, which negatively impacts the overall performance of the device.

Following the establishment of the precise input values for the parameters, TCAD Sentaurus software is employed to simulate each device. This process yields the IdVg curve, IdVd curve, and breakdown voltage (BV) curve for each device. Additionally, Figure 4 shows a double pulsed test circuit to check the reverse recovery and switching characteristics of the device. The external power supply voltage is 600 V, the gate resistance is 4 Ω, the SBD-MOSFET gate source of the upper arm is zero biased and is normally off, and the SBD-MOSFET of the lower arm is controlled by the gate pulse signal to turn on and off. When the SBD-MOSFET on the lower arm is turned off, the SBD on the upper arm is turned on to provide a freewheeling channel, and when the SBD-MOSFET on the lower arm is turned on, the SBD undergoes a reverse recovery process, corresponding to the switching characteristics of the SBD-MOSFET. Through this test circuit, the switching power dissipation of the device can be obtained.

Subsequently, the derived curves are utilized to calculate the threshold voltage, specific on-resistance, breakdown voltage, and switching power dissipation of the device. This process culminates in the formation of a dataset comprising 625 sample groups, thereby enhancing the sample size to effectively bolster the model’s generalization capability. The resulting dataset is partitioned into a training set, a validation set, and a test set in a ratio of 3:1:1. Furthermore, the input and output data in the training set are standardized and normalized. To provide a more intuitive demonstration of the neural network’s generalization, the algorithm is not provided with any information from the test set; instead, the variance and mean of the test set are derived from the raw data of the training set.

## 3. Results and Discussion

### 3.1. Establishment of Neural Network Structure

The design methodology based on neural networks for SBD-MOSFET is comparatively more straightforward than traditional design approaches, as illustrated in the flowcharts presented in Figure 5. It is important to emphasize that, to guarantee the efficacy of the predictive methodology and the accuracy of the outcomes, it is frequently essential to begin with the foundational principles of neural networks, optimize these networks, and enhance their alignment with the data.

To better extract data features, this paper uses a convolutional neural network [28]. Figure 6 presents the architecture of the neural network alongside its configuration prior to integration into the branch network. Due to the intricate nature of SBD-MOSFET devices, the network incorporates multiple convolutional layers to fully leverage the potential information contained within the input data. Nonetheless, the deepening of the network may result in issues such as the vanishing or exploding gradients, which complicates the convergence of neural networks during the training process. Consequently, a dual-branch convolution module has been specifically designed for this study.

The proposed network architecture, grounded in deep learning principles, contains four different modules: the input layer, feature expansion module, feature extraction module, and output layer. In the input layer, the variables W_jfet_, P_wdepth_, W_sbd_, and P_wdopmax_ are designated as inputs, resulting in a total of four input neurons. The feature expansion module is composed of a fully connected layer and a transposed convolutional layer. Since the dimension of the ground input vector of the dataset is small, a fully connected module [29] was added after the input layer for dimension expansion to facilitate subsequent convolution operations. The fully connected layer consists of three layers, which expand the four input neurons to 320, while incorporating a batch normalization layer to mitigate the risk of overfitting. The transposed convolutional layers serve to further integrate and augment features, thereby increasing the dimensionality of the data.

The feature extraction module is characterized by a dual-branch convolutional structure alongside a convolutional module. The dual-branch convolution module features two pathways: the primary path employs a convolutional layer with a stride of 3, while the secondary path utilizes a convolutional layer with a stride of 5. The primary path is composed of three 1D convolutional layers, each with a filter size of 3, whereas the secondary path includes a convolutional layer with a kernel size of 5. By amalgamating the convolutional outputs from both paths, a convolution unit with dual-channel properties is established to yield the final output. To address the complexities inherent in deep learning, three dual-branch convolution modules are incorporated into the design. The dual-branch convolution module can extract data features and prevent gradient disappearance or explosion. Following the dual-branch network, three additional convolutional layers are appended to further deepen the architecture.

In the output layer module, the performance specifications pertinent to SBD-MOSFET devices—namely Vth, R_ON,sp_, BV, and E—are selected as the output neurons. Consequently, it is imperative to reduce the dimensionality of the output from the feature extraction module. This is achieved by introducing three fully connected layers that condense the 4352 neurons down to 4 neurons. Thus, the design of the deep learning-based network structure is finalized.

### 3.2. Predicted Results

In this paper, the Pytorch deep learning framework is rewritten in Python2022 based on torch to implement acceleration specifically for the GPU [30]. The framework is easy to use and supports dynamic computing graphs and efficient memory use. Firstly, each training batch’s activation function, ReLu, is calculated [31]. Then, the ADAM [32] optimizer is used to backpropagate the network parameters until the convolutional neural network converges. At this point, we obtain the trained model. The prediction model uses the early stop [33] method to control whether the training is over. When the prediction error of the model on the verification set is not reduced or reaches a certain number of iterations, the training breaks, and the parameters in the previous iteration results are used as the final parameters of the model. The last saved network weight parameters are taken as the final model parameters. After training, the mean square error (MSE) is used to characterize the prediction effect. It is defined as
(1)$$MSE=\frac{1}{N}\sum_{i=1}^{N}{\left(y_{i}-f_{i}\right)}^{2}$$
where *y_i_* and *f_i_* represent the predicted value and the true value, respectively.

Figure 7 presents a comparative analysis of the predicted values versus the actual values generated by the designed convolutional neural network model. This analysis was conducted over 124 validation rounds, resulting in the generation of four distinct charts. In the figure, the black dots represent the real value, while the red dots represent the predicted value. From the close distance between the red and black dots and the same change trend, it can be seen that the accuracy of prediction reaches a higher level when the sample set is 625 groups.

The graph illustrates a strong correlation between the predictions made by the neural network and the actual values for the four performance metrics of the device, namely threshold voltage (Vth), specific on-resistance (R_ON,sp_), breakdown voltage (BV), and power dissipation (E). During the training and validation phases of the convolutional neural network, the logarithm of the device’s P_wdopmax_ is utilized, which preserves the inherent characteristics of the original data while ensuring effective training of the neural network. The mean square error (MSE) values for Vth, R_ON,sp_, and BV are recorded at 0.0051, 0.0031, and 0.0065, respectively, whereas the MSE for power dissipation (E) is slightly higher at 0.0220. Throughout the training process, the MSE serves as the loss function, quantifying the discrepancy between the predicted and actual values, thereby validating the accuracy of the neural network design.

As illustrated in Figure 8, the error distribution for four performance indicators was analyzed during the verification process. The abscissa represents the MSE value. Specifically, it was found that 98.4% of the errors in the specific on-resistance fell within the range of 0 to 0.1, while 99.2% of the threshold voltage errors were within 0 to 0.05. Furthermore, 100% of the breakdown voltage errors were also confined to the range of 0 to 0.05, and 91.1% of the errors related to switching power dissipation were within 0 to 0.1. This error distribution further substantiates the validity of the neural network employed in the analysis.

Figure 9 illustrates the training loss throughout the training process, with panels (a), (b), (c), and (d) representing the training losses for R_ON,sp_, Vth, BV, and E, respectively. The data presented in the figure indicate that the losses associated with the four device characteristics exhibit a progressive decline from elevated levels to lower levels during the training phase, ultimately stabilizing at approximately zero. The four parameters basically converge between groups 20 and 40 of the data, and the error is close to 0. A relatively small training error is achieved through 160 training sets, indicating that 625 sets of data are sufficient to achieve very high prediction accuracy.

In addition, the characteristic simulation of 625 device configurations using TCAD software consumed approximately 776 h. In contrast, the time required for characteristic prediction using a neural network was about 2 min, representing a reduction of approximately 99.995%. Experimental results indicate that the method of performance prediction for SBD-MOSFETs using convolutional neural networks significantly accelerates the performance research of SiC MOSFET devices, demonstrating considerable potential for application.

In addition, a decision tree (DT) [34], K-nearest neighbor (KNN) [35], random forest (RF), ridge regression (RR), support vector machine (SVM) [36], and deep neural network (DNN) [37] were used to train and predict text data samples and compare them with the predicted results of convolutional neural network, and we used the mean absolute percentage error (MAPE) as the evaluation index of the prediction result. The MAPE is defined as:(2)$$MAPE=\frac{100\%}{n}\sum_{i=1}^{n}\left|\frac{\left(y_{i}-f_{i}\right)}{y_{i}}\right|$$
where *y_i_* and *f_i_* represent the true value and the predicted value, respectively.

Figure 10 shows the MAPE of the prediction effect of four traditional machine learning methods, deep neural networks, and convolutional neural networks. It can be seen that the prediction results of the CNN model are significantly better than other machine learning methods and deep neural networks.

Although the sample data only comprises 625 groups, which is kept in a small order, it has achieved a relatively high prediction accuracy, which shows the correctness of simulation and prediction, and we will also expand the sample set in follow-up research, improving and clarifying the feasibility of the method again.

## 4. Discussion and Conclusions

This study focuses on the performance prediction of neural network applications in the context of SBD-MOSFETs. In this research, a total of 625 sets of device data were acquired through TCAD software simulations, encompassing both device structural parameters and performance metrics. The results indicate a strong correlation between the predicted values and the actual measurements, with mean square deviations for threshold voltage, breakdown voltage, specific on-resistance, and switching power dissipation recorded at 0.0051, 0.0031, 0.0065, and 0.0220, respectively.

This approach significantly minimizes the labor, material resources, and time typically required when utilizing simulation software such as Silvaco or Sentaurus TCAD for device performance analysis. The simulation of the SBD-MOSFET presented in this paper specifically reduced the time by 99.995%. Meanwhile, in the comparison of convolutional neural networks and machine learning, the CNN’s accuracy is much higher than that of the machine learning methods. Overall, the neural network effectively establishes a rapid relationship between device parameters and performance, thereby facilitating researchers in conducting device optimization studies and expediting advancements in the performance prediction of SBD-MOSFETs.

## Figures and Tables

**Figure 1 micromachines-16-00055-f001:**
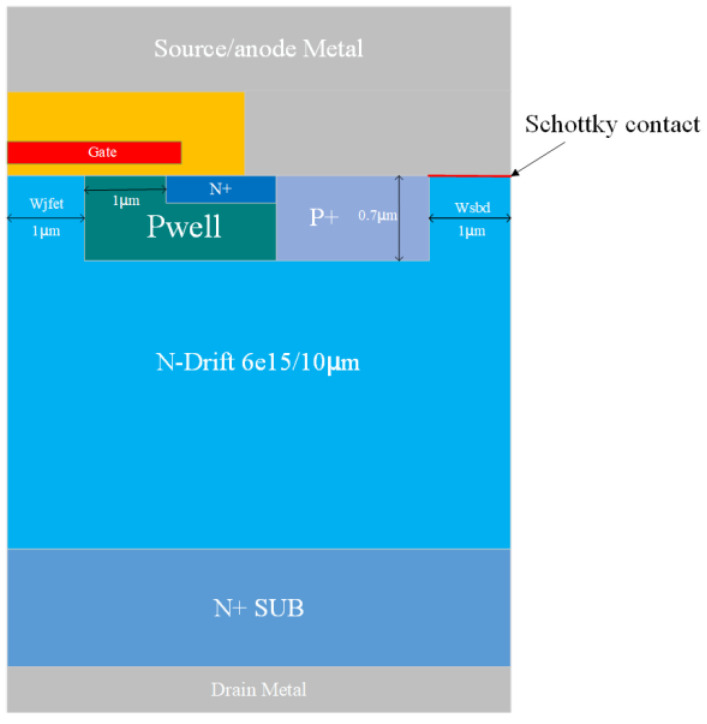
Schematics of SBD-MOSFET.

**Figure 2 micromachines-16-00055-f002:**
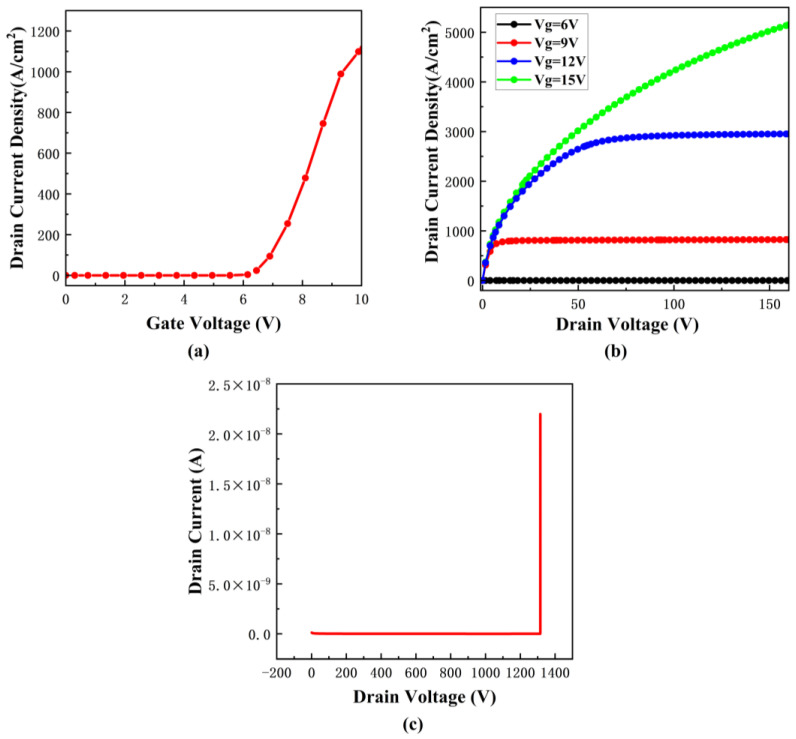
Basic characteristics of the SBD-MOSFET. (**a**) IdVg curve. (**b**) IdVd curve. (**c**) BV.

**Figure 3 micromachines-16-00055-f003:**
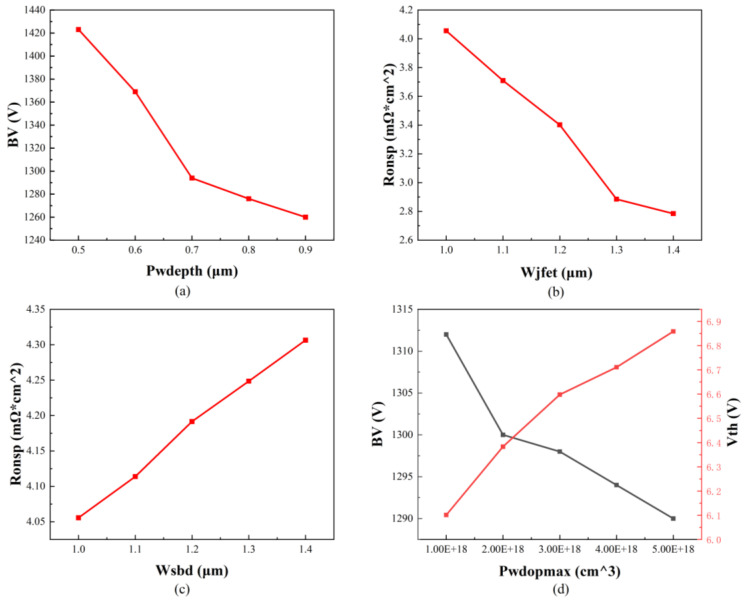
The relationship between the input and output parameters. (**a**) The relationship between the P-well depth and the breakdown voltage. (**b**) The relationship between W_jfet_ and the specific on-resistance. (**c**) The relationship between W_sbd_ and the specific on-resistance. (**d**) The relationship between P-well doping, the threshold voltage, and the breakdown voltage.

**Figure 4 micromachines-16-00055-f004:**
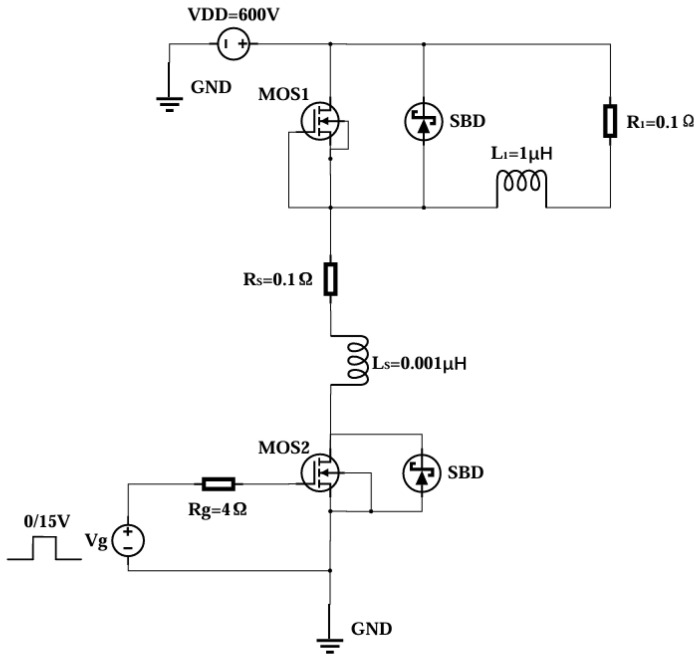
Double pulsed test circuit diagram.

**Figure 5 micromachines-16-00055-f005:**
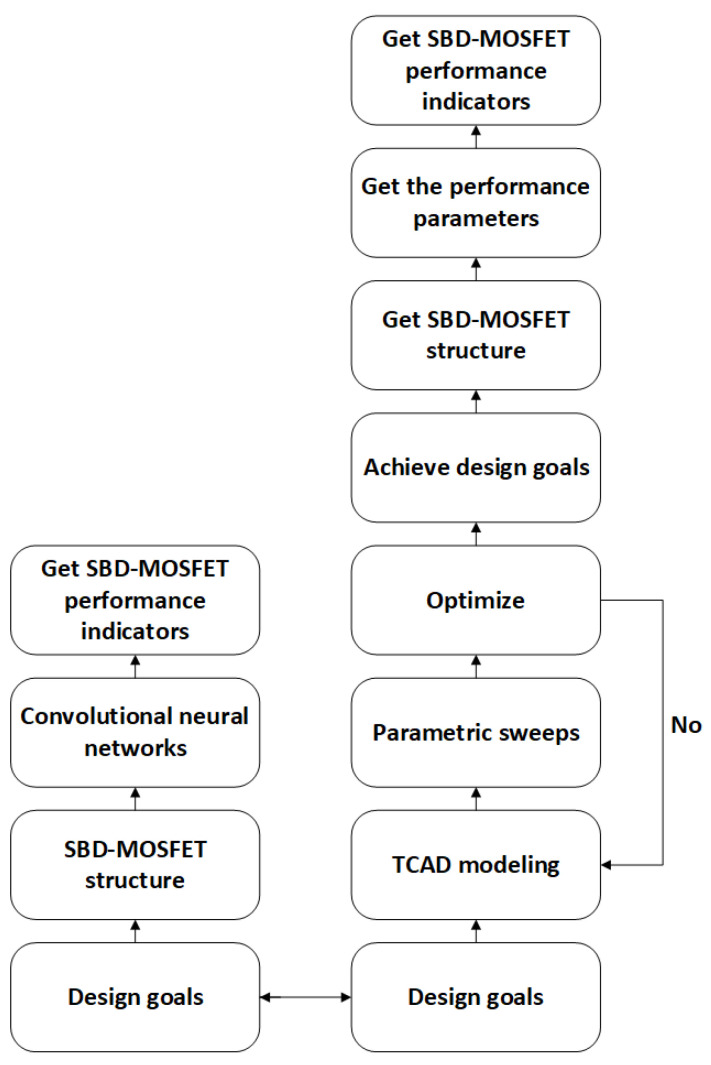
Flowchart of the traditional and neural network-based SBD-MOSFET design approaches.

**Figure 6 micromachines-16-00055-f006:**
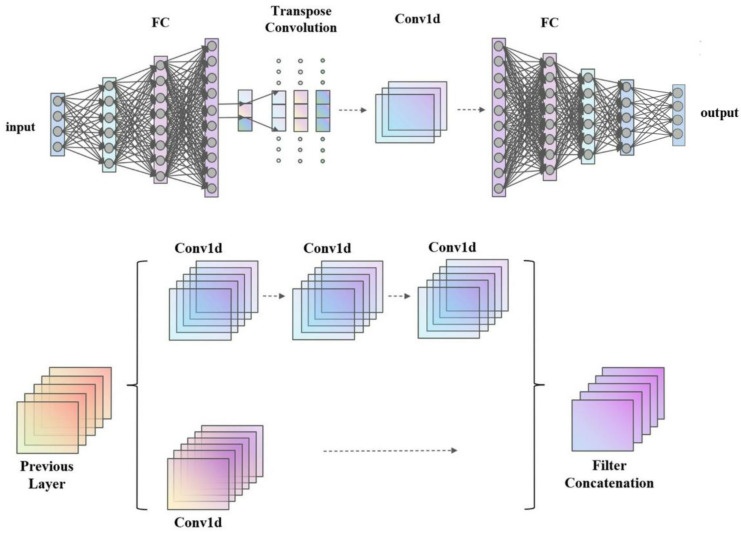
Design of network structure based on deep learning.

**Figure 7 micromachines-16-00055-f007:**
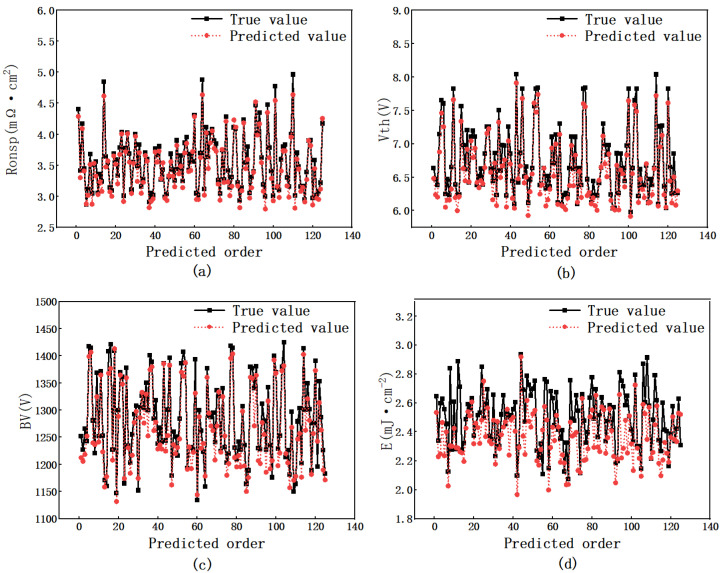
Comparison of predicted and true values of a convolutional neural network model. (**a**) R_ON,sp_. (**b**) Vth. (**c**) BV. (**d**) E.

**Figure 8 micromachines-16-00055-f008:**
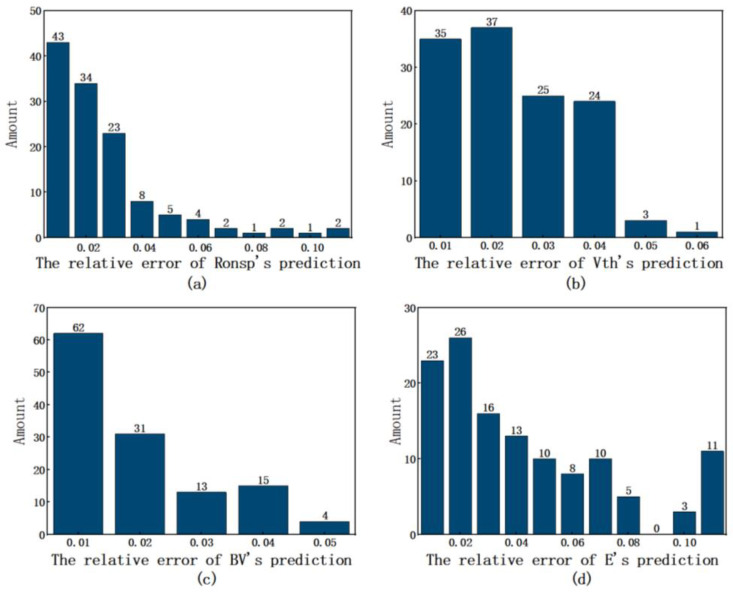
The error distribution histogram of the predicted results. (**a**) R_ON,sp_. (**b**) Vth. (**c**) BV. (**d**) E.

**Figure 9 micromachines-16-00055-f009:**
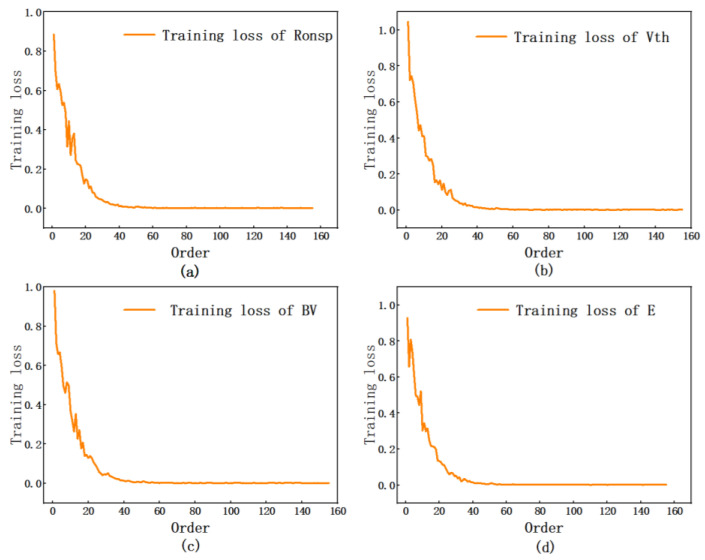
Training loss in the training process. (**a**) R_ON,sp_. (**b**) Vth. (**c**) BV. (**d**) E.

**Figure 10 micromachines-16-00055-f010:**
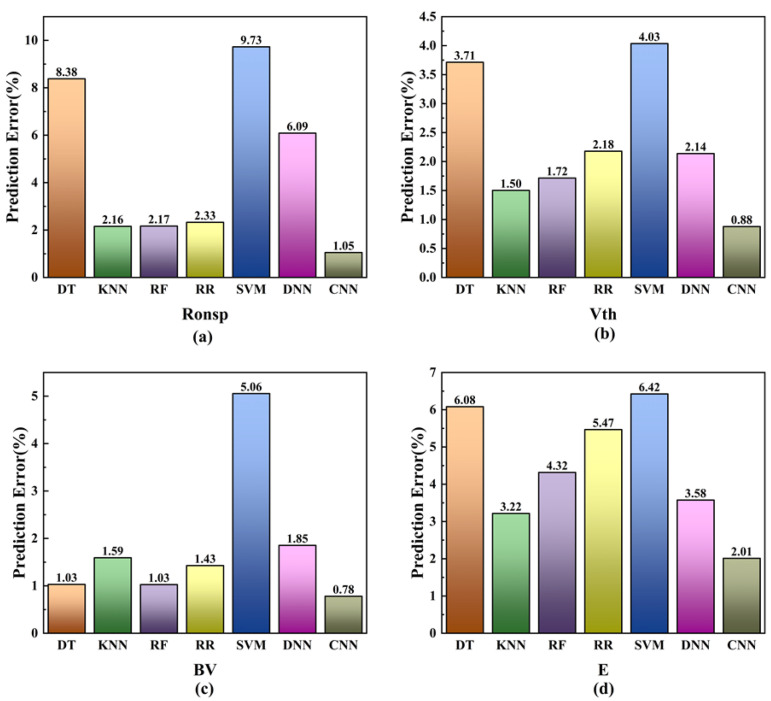
A comparison of the MAPE of a convolutional neural network with conventional machine learning and a deep neural network, where (**a**) is the validation result of R_ON,sp_, (**b**) is the validation result of Vth, (**c**) is the validation result of BV, and (**d**) is the validation result of E.

**Table 1 micromachines-16-00055-t001:** Device parameters.

Paraments	Values
P+ Depth	0.7 μm
P+ Doping	2 × 10^18^ cm^−3^
P+ Width	1.5 μm
N+ Depth	0.2 μm
N+ Doping	2 × 10^19^/4 × 10^17^ cm^−3^
N+ Width	1 μm
Pwell Depth	0.7 μm
Pwell Doping	1 × 10^18^/1 × 10^17^ cm^−3^
Channel Length	1 μm
Tox	0.05 μm
Wsbd	1 μm
Ndrift Thickness	10 μm
Ndrift Doping	6 × 10^15^ cm^−3^
Jfet Width	1 μm
Substrate Thickness	0.2 μm
Substrate Doping	1 × 10^19^ cm^−3^

**Table 2 micromachines-16-00055-t002:** Values of input parameters.

Input Parameters	Values
P_wdopmax_ (cm^−3^)	1 × 10^18^ 2 × 10^18^ 3 × 10^18^ 4 × 10^18^ 5 × 10^18^
P_wdepth_ (μm)	0.5 0.6 0.7 0.8 0.9
W_jfet_ (μm)	1 1.1 1.2 1.3 1.4
W_sbd_ (μm)	1 1.1 1.2 1.3 1.4

## Data Availability

The original contributions presented in this study are included in the article. Further inquiries can be directed to the corresponding author.

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
