# Peer review of "SiC MOSFET with Integrated SBD Device Performance Prediction Method Based on Neural Network"

_micromachines, 2024, doi:10.3390/mi16010055_

Round 1
Reviewer 1 Report
Comments and Suggestions for Authors
The authors used CNN based neural network algorithms to 'predict' the simulation results of SBD-SiC MOSFET. Similar approaches have been reported in semiconductor device/material modelling and simulation, the results are not always convincing or proven useful.
(1) The authors need to summarize the recent advances in this topic, not limited to SBD SiC MOSFET, in the introduction, and highlight the shortcomings and how to avoid them in this research.
(2) The authors chose CNN without detailed justification against other NN topologies, nor with any results to demonstrate that this is the most suitable NN.
(3) 625 sets of device data are rather limited, and therefore the extremely high prediction accuracy is not convincing. A more thorough evaluation of the methods, including the limitations on when and how the accuracy starts to converge, improve, and optimise are needed.
(4) The so-called 'predication' is based the physical modelling process simulated with the Silvaco or Sentaurus TCAD tools. The neural network 'prediction' does not involve the physics and can be considered merely as multi-dimensional extrapolation of TCAD results. Please discuss in detail the limitation of this kind of 'prediction'.
Author Response
Thank you very much for taking the time to review our manuscript(micromachines-3334892) entitled “SiC MOSFET with integrated SBD Device Performance Prediction Method Based on Neural Network”. Please find the detailed responses below and the corresponding revisions/corrections highlighted/in track changes in the re-submitted files. Edits in the original text are highlighted in the reply in blue slanted font. Corrections to the original text are highlighted in red font and underlined

Reviewer 2 Report
Comments and Suggestions for Authors
This manuscript is based on data generated from TCAD simulations and employs neural networks to predict the static and dynamic performance parameters of SBD-MOSFETs. However, the manuscript suffers from numerous fundamental issues and requires substantial revisions before it can be considered for publication.
Comment 1:
On page 3, the authors mention that the TCAD simulation results are consistent with the characteristics of devices reported in the literature. However, no specific references or comparative data are provided. This is a critical issue, as the accuracy of TCAD simulations directly impacts the reliability of the predictive model. The authors must provide either a detailed comparison between the TCAD simulation results and measured data from the literature or include appropriate references to support their claim.
Comment 2:
On page 8, line 204, the manuscript incorrectly describes the ReLU function as a loss function. ReLU is an activation function and cannot quantify the discrepancy between predicted and target values. This is a fundamental error that should not occur in a manuscript of this caliber. The authors must explicitly state the correct loss function used in their model.
Comment 3:
The authors employ a convolutional neural network (CNN) for this prediction task. Given the dimensions of the problem, such a complex model seems unnecessary. Furthermore, the prediction process still requires two minutes, which severely limits its practical application. The authors should compare the CNN with a more classical deep neural network (DNN) [R1] while ensuring that the parameter counts of both models are equivalent. This would provide a better justification for using the CNN.
Comment 4:
In Figure 6, the CNN's output layer consists of five neurons, but the manuscript specifies that only four performance parameters are predicted. This inconsistency must be addressed and clarified.
Comment 5:
The manuscript's references are problematic in both completeness and relevance:
l References [14] and [20] are incomplete.
l Reference [15] is not related to machine learning modeling, yet it is cited when describing the loss function as ReLU.
Comment 6:
In Figure 8, the x-axis label is ambiguous, does it represent relative error or mean squared error (MSE)? The authors should clarify this in both the figure caption and the main text.
Comment 7:
The manuscript contains several formatting errors and inconsistencies that detract from its overall quality:
l Abbreviations such as "BV" and "Vth" should be defined with their full names upon first mention, along with the corresponding abbreviations.
l Subscripts are inconsistently formatted: for instance, "Vth" uses a subscript for "th," while "Pwdopmax" does not. A consistent style must be adopted throughout the manuscript.
l Units in figures and tables are either inconsistent or missing:
n In Figure 3(c), the y-axis unit is "cm^2," whereas in Figure 4(a), it is "cm2." These should be unified.
n Figure 4 lacks units for inductance.
n Tables 1 and 2 do not specify the units of the structural parameters. These must be clearly indicated.
l On page 10, lines 243 and 244 are not continuous.
Reference:
1. Lee, J.K.; Ko, K.; Shin, H. Prediction of Random Grain Boundary Variation Effect of 3-D NAND Flash Memory Using a Machine Learning Approach. IEEE Trans. Electron Devices 2022, 69, 447–449, doi:10.1109/TED.2021.3130858.
Author Response

(The authors gave the same response as above.)

Reviewer 3 Report
Comments and Suggestions for Authors This research work introduces a novel approach utilizing neural network machine learning 14 to predict the static and dynamic characteristics of SBD-MOSFET. In this research, SBD-MOSFET 15 devices are modeled and simulated using Sentaurus TCAD software, resulting in the generation of 16 625 sets of device structure and sample data, which serve as sample set for the neural network. The results of this work are promising, but there are a few comments: 1. The threshold voltage, as derived from the Id vs Vg curve in Figure 2(a), is measured at 6.101V, which is higher than the value reported in the reference literature. This difference can be attributed to the implementation of the 0.7μm P-well depth. Also as shown in Figure 2(c), this change leads to a reduction in the device's breakdown voltage, which is determined to be 1315V from the Breakdown Voltage graph, falling below the reference value. Why does including a 0.7μm P-well in the SBD-MOSFET device model bring this increase in gate voltage and reduction in Breakdown Voltage ?2. Some more refrences may be added to further explore applications area of SiC, for example https://doi.org/10.1116/1.4884756 3. On what basis are the values of the device parameters selected in Table 1 in the TCAD model for the SBD-MOSFET and the input parameters in Table 2 ?
4. Please modify the abstract and conclusion sections accordingly.
Author Response

(The authors gave the same response as above.)

Round 2
Reviewer 1 Report
Comments and Suggestions for Authors
The authors have addressed my comments appropriately.